# License Plate Image Reconstruction Based on Generative Adversarial Networks

**Mianfen Lin [1], Liangxin Liu [1], Fei Wang [1], Jingcong Li [1] and Jiahui Pan [1,2,*]**

[1] School of Software, South China Normal University, Foshan 528225, China; linmf@m.scnu.edu.cn (M.L.); lix0906s@m.scnu.edu.cn (L.L.); fwang@scnu.edu.cn (F.W.); jcli@m.scnu.edu.cn (J.L.)
[2] Pazhou Lab, Guangzhou 510330, China
[*] Correspondence: panjiahui@m.scnu.edu.cn

**Abstract:** License plate image reconstruction plays an important role in Intelligent Transportation Systems. In this paper, a super-resolution image reconstruction method based on Generative Adversarial Networks (GAN) is proposed. The proposed method mainly consists of four parts: (1) pretreatment for the input image; (2) image features extraction using residual dense network; (3) introduction of progressive sampling, which can provide larger receptive field and more information details; (4) discriminator based on markovian discriminator (PatchGAN) can make a more accurate judgment, which guides the generator to reconstruct images with higher quality and details. Regarding the Chinese City Parking Dataset (CCPD) dataset, compared with the current better algorithm, the experiment results prove that our model has a higher peak signal-to-noise ratio (PSNR) and structural similarity (SSIM) and less reconstruction time, which verifies the feasibility of our approach.

**Keywords:** super-resolution reconstruction; generative adversarial network; residual dense network; python

## 1. Introduction

With the increase in population and expanding economic scale, in recent years, car usage has been increasing, increasing the pressure on the transportation system. As an emerging technology, license plate recognition is a significant component of an intelligent transportation system. It is widely used in such fields as vehicle positioning, vehicle identification at highway toll stations and speed measurement. However, under complex environmental conditions, such as occlusion, shadows, light changes and bad weather conditions, the quality of the license plate pictures taken is often not satisfactory, which also makes the subsequent processing more challenging. Therefore, taking full advantage of deep learning technology to repair and reconstruct blurry license plate images, it is convenient to perform subsequent positioning or other operations on the captured vehicles. This is realistic for maintaining social traffic safety and realizing traffic automation management.

In addition, the algorithm can also be applied to monitoring devices, satellite images and medical images. For example, in the field of traffic monitoring, the police need the details of the image in order to recognize the accurate license plate number and make accurate judgments. In the field of medical images, doctors need the details of images to make accurate judgments.

The purpose of this paper is to repair and reconstruct the blurry license plate images of lower quality. In this study, a super-resolution reconstruction algorithm based on Generative Adversarial Networks (GAN) is proposed. The significance of the algorithm lies in: (1) The introduction of the residual dense network, which is conducive to the model to retain more high-frequency detail information of the image, while the batch normalization (BN) layer is removed. Experiments have proven that it is beneficial to

model training. (2) The introduction of progressive upsampling, which is conducive to generating high-resolution images.

This paper consists of five sections. In Section 1, we highlight the research background of image reconstruction algorithms. In Section 2, the principle of GAN, the network model of this paper and the reasons for using GAN are presented. In Section 3, the network model and principle used are introduced specifically. In Section 4, the model is presented and the experimental results are analyzed, as well as the shortcomings. In Section 5, a summary is included.

*Related Work*

At present, the reconstruction of fuzzy images can be broadly divided into three main categories: interpolation-based image reconstruction method [1], image reconstruction method based on reconstruction [2] and deep learning-based image reconstruction method [3]. As the earliest proposed image reconstruction method, the image reconstruction method based on interpolation has the advantages of simplicity and speed. However, this method will cause artificial effects such as sawtooth and ringing, so it has been quickly abandoned. The method based on reconstruction is proposed for the shortcomings of the interpolation method. However, the algorithm is complicated in operation and low in accuracy, thus it is not worth further study for scholars.

Image super-resolution (SR) reconstruction based on deep learning, as an image processing technology that can enrich image details and restore distorted images with higher quality, has attracted widespread attention from domestic and foreign researchers since it was proposed. Dong et al. [4] proposed a super-resolution reconstruction method based on a convolutional neural network. Through training, the network can learn the nonlinear mapping relationship between low-resolution and high-resolution images, so as to complete image reconstruction. In the research of Kim et al. [5,6], the convergence rate of the model was improved by increasing the depth of the convolutional network and introducing a residual learning structure. Yulun Zhang's [7] team makes use of residual dense structure, which is conducive to extracting a large number of local features from the densely connected convolutional layers. Ledig [8] applied the GAN framework to the super-resolution reconstruction of images and obtained satisfactory results. The model was named SRGAN, a GAN for image super-resolution (SR). Kui Fu et al. [9] analyzed that GAN has more advantages than a traditional convolutional neural network. In [10], the author changed the fundamental module of GAN into a residual-in-residual dense block (RRDB) and improved the output method of the perceptual loss function. Recently, Chen, Y et al. [11] proposed an image super-resolution reconstruction method based on the feature map attention mechanism, which realized the reconstruction from the original low-resolution image to the multi-scale super-resolution image. Experimental results show that the method can effectively improve the visual effect of the image. In addition, the evaluating indicator of Peak Signal to Noise Ratio (PSNR) and Structural Similarity Index (SSIM) has been improved to a certain extent, and the effect of using feature mapping attention mechanism in image super-resolution reconstruction is useful and effective. In [12], a super-resolution reconstruction method of single frame character image based on wavelet neural network is proposed. The structure and interface of the image acquisition unit of solid-state image sensor are designed. The blur degree of the reconstructed image is always less than 5%, and the accuracy can be maintained at 80%~90%.

## 2. GAN

### 2.1. GAN Principle

GAN is an unsupervised learning model proposed by Ian J. Goodfellow et al. [13] in 2014, which can automatically learn the complex distribution rules among sample data to generate "machine data" that conform to the sample data distribution. GAN adopts the idea of a "zero-sum game" in game theory. The sum of the interests of both parties in the game is a constant, usually 0. The gains of one party necessarily mean the losses of the

other party. The two models of generator and discriminator are regarded as two parties participating in the game. For the generator, learn the real data distribution characteristics to generate "machine data" similar to the real data. As for discriminator, it improves its own discriminative ability through continuous learning and optimization. The ultimate objective is to correctly judge whether the input data are real data or still generated data and eventually output 0 or 1.

The training process of GAN is a Minimax Game problem, in which the discriminator will minimize the error in the process of discriminating true and false data, while the generator will maximize the probability of the discriminator's judgment error. This contradictory training process enables the generator to generate "machine data" similar to the sample data, so as to achieve the effect of mix the spurious with the genuine. The ultimate goal is to achieve Nash equilibrium [14].

The model adopted in this study is inspired by the enhanced super-resolution generative adversarial networks (ESRGAN) model [10]. In ESRGAN, it proposes to remove all BN layers. Meanwhile, it is proposed to replace the original basic block with RRDB, which combines a multi-layer residual network and dense connections. When the single residual network is applied to extract image features, feature loss is hugely serious. The residual dense network can make full use of all the hierarchical features of low-resolution images, some of which are conducive to generating high-resolution images and reconstructing large-scale factor images.

The algorithm based on ESRGAN has a few improvements: (1) The network of this paper replaces the original basic block with RRDB, combined with multi-layer residual network and dense connection, and adopts progressive upsampling. (2) Simultaneously, for the special image of a license plate, we focus on advanced features such as edge and letter direction in order to minimize the mixture of VGG loss (a type of content loss) and mean square error (MSE) as the loss function parameter of our algorithm. (3) Finally, peak signal-to-noise ratio (PSNR) [15] and structural similarity (SSIM) are applied as objective evaluation indicators for the quality of reconstructed images.

### 2.2. Reasons for Using GAN

When the probability distribution of real data is difficult to calculate, for example, the input is various pictures of the real world, traditional generative models cannot be directly applied. However, GAN can still be used. Through the training mechanism of generator and discriminator confrontation, the generator can be used to approximate the probability distribution that is difficult to calculate (generate realistic pictures) [13].

According to the actual results, they seem to produce better samples (sharper and clearer images) than other models.

### 3. Network Structure

Figure 1 shows the overall network architecture of the proposed method.

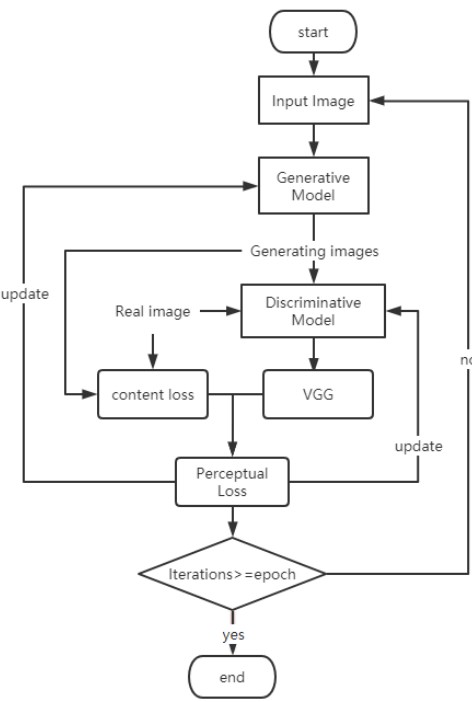

**Figure 1.** Overall Network Architecture.

### 3.1. Generative Model

The structure of the generative network model in this study is shown in Figure 2. In the stage of generating information to extract information, a residual dense network is applied to extract information from images. In the stage of image reconstruction, the progressive upsampling method is adopted. Therefore, in simple terms, our generative model is mainly composed of two parts: residual dense network and progressive sampling.

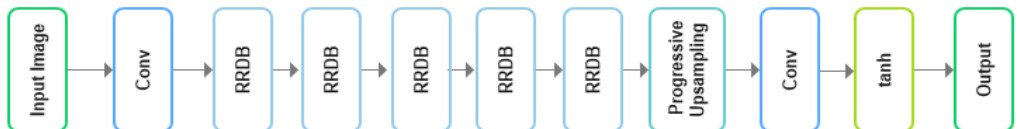

**Figure 2.** Generative Model.

#### 3.1.1. Residual Dense Network

The residual dense network (RDN) is mainly composed of four parts, namely shallow feature extraction network (SFENet), residual dense bock (RDB), dense feature fusion (DFF) and upsampling network (UPNet), where SFENet is composed of a convolution (Conv). The RRDB has a deeper and more complex structure than SRGAN's original residual block (RB). RRDB removes the BN layer. This is because the BN layer tends to introduce bad artifacts when the statistics of the training and test datasets differ considerably. This performance limits its own generalization ability. In particular, RRDB has a residual-in-residual structure, which enables residual learning to be applied to different layers. An RRDB contains three dense blocks (DB). Each dense block contains five convolutions (Conv) and 4 activation functions (leaky rectified linear unit, LReLU), the size of the convolution kernel in each combination is $7 \times 7$, and the step size is 1. They are interchangeably and densely linked to form the dense block.

The residual dense block consists of densely connected layers and local feature fusion with local residual learning. It also supports continuous memory between RRDBs. The output of an RRDB can be directly linked to all the subsequent RRDB layers. The state can be transmitted continuously. Each convolutional layer of RRDB can access all subsequent

layers and transfer information that needs to be retained. The residual dense network combines the characteristics of residual network and dense network, fusing global features and local features to form a continuous memory mechanism. The residual dense network structure is shown in Figure 3.

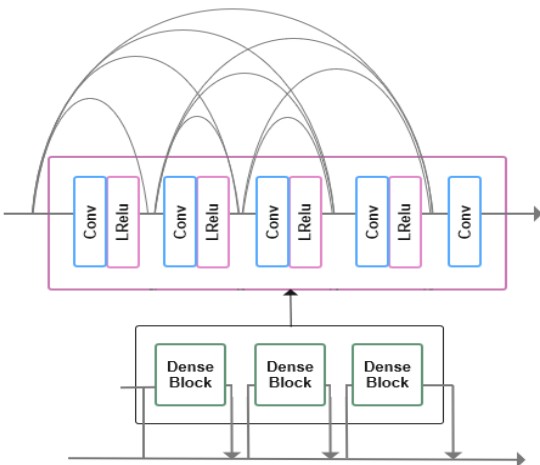

**Figure 3.** Residual Dense Network Structure.

### 3.1.2. Progressive Upsampling

Compared with the ordinary upsampling network, a more advanced progressive upsampling network is adopted. Progressive upsampling is realized by sub-pixel convolution, and its network structure is shown in Figure 4. In the sub-pixel convolution, suppose the length and width of the sub-pixel image (set to h, w), and the sampling factor r, and generate h×w×r×r sub-pixel images through ordinary convolution. Eventually, the sub-pixel images are reorganized in sequence, in order to obtain high-resolution images. Compared with the deconvolution method, the most prominent advantage of sub-pixel convolution is that it has a larger receptive field and can provide more accurate details of context information.

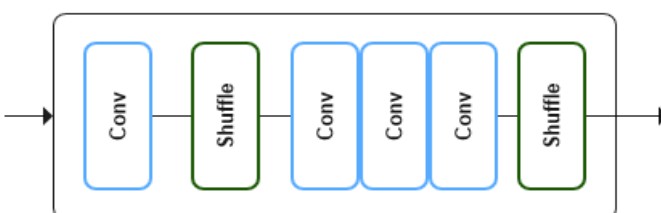

**Figure 4.** Progressive Upsampling Network Structure.

### 3.2. Discriminative Model

PatchGAN [16] maps the input to the patch (matrix) X of N × N, and the value represents the probability that each patch is a true sample. PatchGAN makes a true or false judgment on the image patch, and calculates the mean value of the judgment result, which is the final output of the discriminator.

In this study, the discriminator draws on the idea of PatchGAN [16] and maps the input to an N × N matrix M, where each element $M_{ij}$ corresponds to a specific image block (Patchi,j) in the original image. The average value of the filter network matrix is used as the final judgment result, and the threshold is set to output 0 or 1. The discriminator is designed as a six-layer fully convolutional neural network, each convolutional layer with a convolution kernel size of 4 × 4 and a stride of 2 for the purpose of increasing the size of the receptive field and strengthening the model's ability to explore important information.

Simultaneously, reduce the feature scale by using the global pooling operation to increase the speed of network computing.

Compared with traditional GAN discriminator, the discriminator in this study allows images of any size as input. Its design is more flexible. Moreover, by judging each patch, local image feature extraction is realized, which is more conducive to promoting the generator to rebuild higher quality and more detailed high-resolution images. The model architecture of the discriminator is shown in Figure 5.

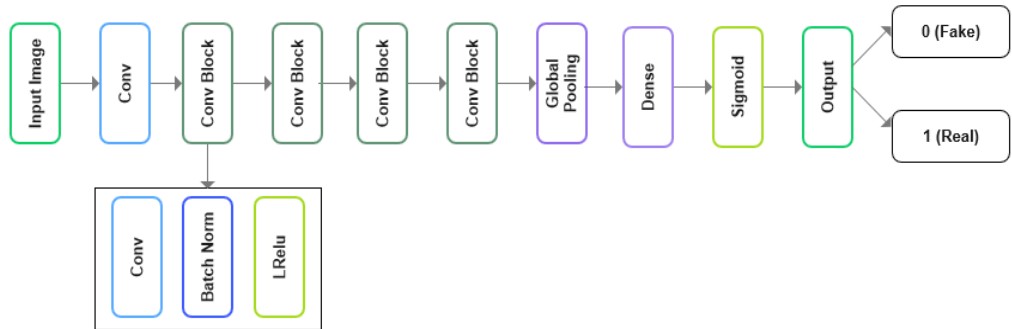

**Figure 5.** Discriminative Model Structure.

### 3.3. Loss Function

We use a mixture of MSE (content loss) and minimized *VGG* as the loss function of the model. The ultimate purpose of the loss function of the GAN network is to make the generator and discriminator reach Nash equilibrium. In ESRGAN's paper, a new content loss function is defined, which calculates the mean square error in the feature space of the *VGG* network. It highlights the characteristics of the original image, rather than the mean square error in the image space. The Euclidean distance between generated image and the original HD image is smaller. The *VGG* loss is defined as follows:

$$l_{VGG/i,j}^{S,R} = \frac{1}{W_{i,j}H_{i,j}} \sum_{x=1}^{W_{i,j}} \sum_{y=1}^{H_{i,j}} \left(\Phi_{i,j}(I^{HR})_{x,y} - \Phi_{i,j}\left(G_{\theta G}\left(I^{LR}\right)\right)_{x,y}\right)^2 \tag{1}$$

where $W_{i,j}$ and $H_{i,j}$ represent the dimensions of each feature map in the *VGG* network, $\Phi_{i,j}$ represents the feature map of the *j*-th convolutional layer before the *i*-th convolutional layer in the network, $I^{HR}$ represents the high-resolution image, *G* represents generation network, $\theta G$ represents generation network parameters, and $I^{LR}$ represents the low-resolution image.

After optimization, the loss function consists of minimizing *VGG* and MSE (content loss). The specific role of content loss is to force the model to pay attention to edges during reconstruction to ensure that the upper skeleton matches the ground truth, which has proved to be extremely beneficial for license plate image reconstruction. The formula is as follows:

$$Loss = 0.01 * \frac{1}{n} \sum_{i=1}^{n} \left|\left|VGG(\hat{Y}) - VGG(Y)\right|\right|_2 + \frac{1}{n} \sum_{i=1}^{n} \left|\left|\hat{Y} - Y\right|\right|_2 \tag{2}$$

where *VGG* is expressed as formula 1, $\hat{Y}$ represents the observed value of image parameters, and *Y* represents the predicted value of image parameters.

## 4. Experimental Results

### 4.1. Experiment

The experiment in this article is performed on a PC with an Intel Xeon E5-2689C2 CPU, an NVIDIA GTX970 16 GB GPU. The operating system is Windows 10, the deep learning framework Tensor Flow 2.1.0, and CUDA 10. 1.

The dataset in this study is the Chinese City Parking Dataset (CCPD) [17]. It contains 200k+ pictures taken under different environmental conditions. However, pictures in the dataset cannot be directly used as the system's input. The pictures should be preprocessed. The preprocessing steps are as follows: (1) Cut out the license plate images from the pictures in the dataset. (2) We should filter out images with poor brightness and contrast and keep only favorable parts as input to the model. After preprocessing and selection, 1000 images were selected as experimental data, among which 800 were training images, 100 were test images, and the rest were verification images. Our experiment chose the Adam algorithm as the network optimizer; the learning rate was initialized to $1 \times 10^{-5}$, and adjusted along with the training process. The best learning rate was found to be $1 \times 10^{-3}$; the batch size (batch size) was set to 16; epochs were set to 100. The weight is initialized to values conforming to the standard deviation of 0.01 based on normal distribution.

In order to verify the effectiveness of the algorithm, this paper adopts PSNR and SSIM as the objective evaluation indicators of the quality of reconstructed images. PSNR is specifically defined as the ratio of the maximum possible power of a signal to the destructive noise power that affects its representation accuracy. It is a measure of image quality. The larger the PSNR, the better the image quality. SSIM compares the similarity of two images in three dimensions: brightness, contrast and structure. The closer the value of SSIM is to 1, the better the reconstruction strength.

PSNR formula:

$$f_{PSNR} = 10\lg\left(\frac{(2^8 - 1)^2}{f_{MSE}(y, y')}\right) \tag{3}$$

$$f_{MSE} = \frac{1}{rc} \sum_{i=0}^{r-1} \sum_{j=0}^{c-1} \left(y_{i,j} - y'_{i,j}\right) \tag{4}$$

SSIM formula:

$$f_{SSIM} = \frac{\left(2\mu_y \mu_{y'} + C_1\right)\left(\sigma_{y'y} + C_2\right)}{\left(\mu_y^2 + \mu_{y'}^2 + C_1\right)\left(\sigma_y^2 + \sigma_{y'}^2 + C_2\right)} \tag{5}$$

### 4.2. Experimental Result

First of all, in order to verify the effectiveness of the algorithm in this paper, the algorithm is compared with the current more advanced image reconstruction techniques such as Bicubic [18], SRCNN [19], SRGAN [8], ESPCN [20]. Comparison of PSNR and SSIM. The results are shown in Tables 1 and 2.

**Table 1.** Comparison of PSNR and SSIM values between different SR algorithms in image reconstruction.

| Dataset | Evaluating Indicator | Bicubic | SRCNN | ESPCN | SRGAN | Our Algorithm |
|---------|---------------------|---------|-------|-------|-------|---------------|
| CCPD | PSNR | 22.45 | 24.75 | 28.70 | 26.37 | 26.08 |
|  | SSIM | 0.71 | 0.8 | 0.76 | 0.79 | 0.77 |

**Table 2.** Comparison of typical image experiment results.

| Figure | Bicubic | | SRCNN | | ESPCN | | SRGAN | | Our Algorithm | |
|--------|---------|------|-------|------|-------|------|-------|------|---------|------|
|  | PSNR | SSIM | PSNR | SSIM | PSNR | SSIM | PSNR | SSIM | PSNR | SSIM |
| 6 | 23.72 | 0.75 | 20.06 | 0.82 | 27.06 | 0.82 | 25.92 | 0.87 | 26.12 | 0.67 |
| 7 | 25.53 | 0.84 | 24.16 | 0.65 | 25.73 | 0.87 | 23.09 | 0.88 | 24.67 | 0.78 |
| 8 | 24.25 | 0.85 | 22.02 | 0.61 | 28.75 | 0.85 | 28.55 | 0.87 | 26.89 | 0.87 |
| 9 | 22.57 | 0.79 | 21.58 | 0.75 | 28.16 | 0.69 | 26.55 | 0.89 | 27.08 | 0.64 |

Table 1 reveals that the SR algorithm based on deep learning is obviously superior to the traditional Bicubic algorithm, and the GAN-based algorithm can obtain higher PSNR and SSIM values than the convolutional neural networks (CNN) based algorithm, which has a better image reconstruction strength. Compared with SRCNN and Bicubic, PSNR and SSIM values of the algorithm in this study were significantly improved. Compared with SRGAN and ESPCN, the algorithm presented in this paper was not improved significantly. Therefore, there is still some research space for our model, which needs to be further improved.

It can be seen from Table 2 that for some typical image reconstructions, such as the reconstruction of Figures 6–9, compared with SRCNN and Bicubic, the PSNR and SSIM values of the model algorithm in this paper are considerably improved. The average PSNR value of the algorithm in this paper is improved by about 2.0~4.0 dB, and the SSIM value is improved by about 0.01~0.06. Compared with ESPCN and SRGAN, the PSNR and SSIM values of the algorithm model in this paper are not significantly improved. Even in a typical picture, the PSNR and SSIM values are still low. This is obviously a fly in the ointment, which requires us to make subsequent improvements to the model.

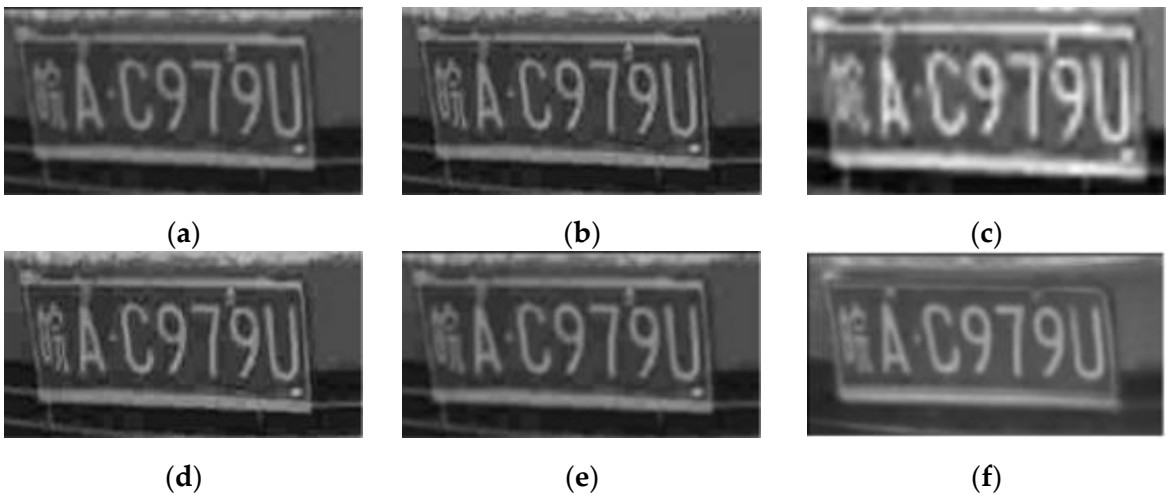

**Figure 6.** (**a**) Ground Truth (**b**) Bicubic; (**c**) SRCNN; (**d**) ESPCN; (**e**) SRGAN; (**f**) Our Algorithm.

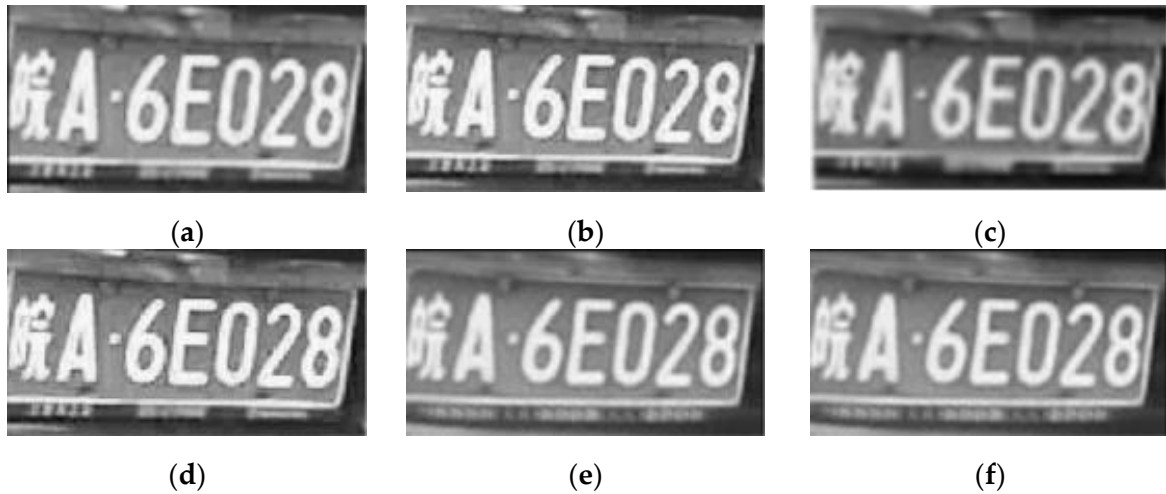

**Figure 7.** (**a**) Ground Truth (**b**) Bicubic; (**c**) SRCNN; (**d**) ESPCN; (**e**) SRGAN; (**f**) Our Algorithm.

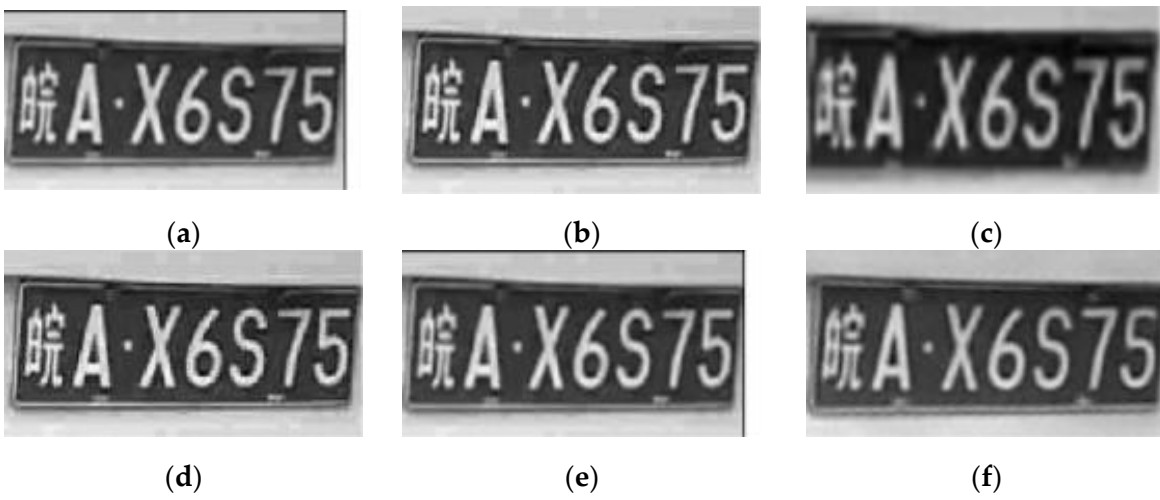

**Figure 8.** (**a**) Ground Truth (**b**) Bicubic; (**c**) SRCNN; (**d**) ESPCN; (**e**) SRGAN; (**f**) Our algorithm.

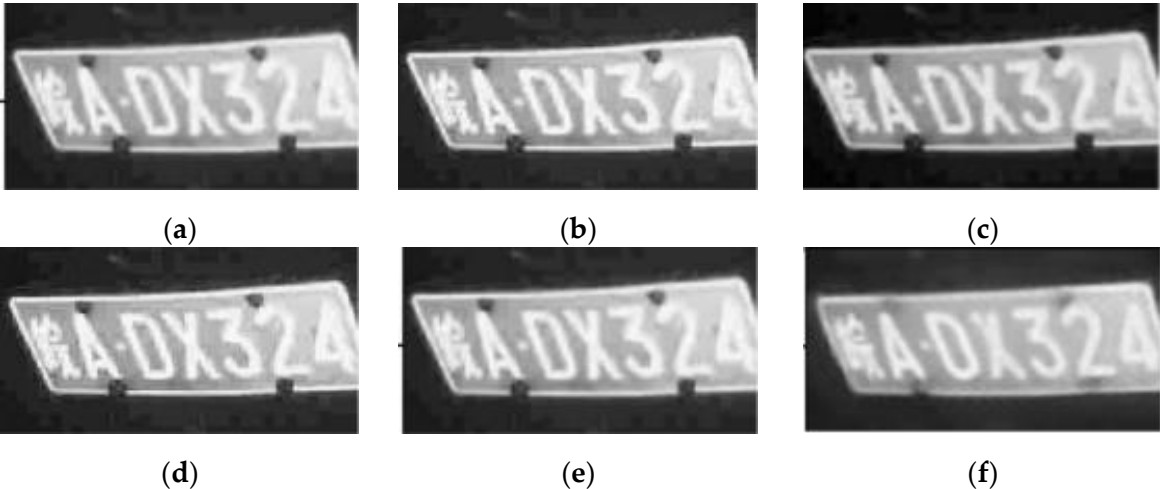

**Figure 9.** (**a**) Ground Truth (**b**) Bicubic; (**c**) SRCNN; (**d**) ESPCN; (**e**) SRGAN; (**f**) Our algorithm.

In addition, the superiority of the model is verified in terms of PSNR and SSIM. The running efficiency of the algorithm is tested. Table 3 reveals that the Bicubic algorithm takes the shortest time, which is related to its simple interpolation method. Moreover, the average time-consuming of the algorithm based on the CNN framework is relatively high. This is because CNN contains a large number of convolution calculations, resulting in more time-consuming image reconstruction. Moreover, among a variety of algorithms based on the GAN framework, this algorithm has the highest average data processing speed. Compared with the least time-consuming SRCNN in the convolutional neural network, our algorithm shortened about 0.03 s, while the PSNR value increased by 2.3 dB. The experimental results show that the algorithm consumes less running time under the premise of ensuring the quality of the reconstructed image, but has favorable running efficiency, which is very satisfactory.

**Table 3.** Comparison of the running time of each algorithm.

| Algorithm | Bicubic | SRCNN | ESPCN | SRGAN | Our Algorithm |
|-----------|---------|-------|-------|-------|---------------|
| Time | 0.023 | 0.063 | 0.068 | 0.089 | 0.060 |

## 5. Discussion

It is proposed that a high-resolution reconstruction algorithm for license plates based on GAN can help machines better recognize license plates. Compared with the existing methods, the Generative Adversarial Networks (GAN) can obtain higher PSNR and SSIM values. Regarding the CCPD dataset, our PSNR and SSIM reached 26.80 and 0.77.

In reality, the speed of license plate image recognition is often an important reference index that cannot be ignored. In Table 3, it can be seen that while maintaining a good reconstruction effect, the algorithm presented in this paper has less running time, which has an obvious speed advantage for application in license plate recognition. This is undoubtedly a compromise scheme with a better reconstruction effect and lower time consumption.

The residual dense network is composed of a residual network and dense network. Combined with the advantages of both, it is convenient to obtain the image features extracted from each network layer to retain more high-frequency information of the image. Simultaneously, a progressive upsampling method is used to improve the super-resolution reconstruction effect under a large scaling factor. Meanwhile, in the discriminator model, PatchGAN is used for reference, and the discriminator is designed into a 6-layer full convolutional neural network, which can increase the size of the receptive field and enrich the details of the reconstructed image.

Finally, the algorithm in this paper is evaluated from multiple dimensions. We use PSNR and SSIM as the evaluation indicators. According to Tables 1 and 2, our algorithm value reached the average value of the current mainstream algorithm. However, there is still a big gap with the current top algorithm model. Through preliminary analysis, it is believed that the discriminator of the model is designed with a small number of layers and cannot distinguish machine data from real data in an efficient manner. This will be the focus of our next research.

## 6. Conclusions

The application of deep learning to image reconstruction is not long-lasting, but the speed of development is very fast, and the effect of the repaired image is better than the traditional technique. Based on the GAN, this paper introduces the residual dense network and the progressive method, and achieves remarkable results in super-resolution reconstruction. The algorithm in this paper takes dense residual network as the core module for feature extraction, strengthens the propagation of features, and adopts progressive upsampling to reconstruct large-scale factor images. Simultaneously, the algorithm uses a PatchGAN-based discriminator to make more accurate judgments, so as to guide the generator to reconstruct images with higher quality and high details. The experimental session shows that compared with other SR algorithms, the algorithm in this paper reduces the learning difficulty of license plate image reconstruction. Without introducing too much space and time cost, the reconstructed image has higher PSNR, SSIM values and fewer Edge artifacts. The focus of future research work, on the one hand, is to further optimize the network structure and reduce the difficulty of training, and on the other hand, it is to study how to avoid the backdoor attack [21] and enhance the robustness of the network model, so that the license plate reconstruction algorithm can be used in real life as soon as possible.

**Author Contributions:** Conceptualization, M.L. and J.P.; methodology, L.L. and J.L.; investigation, J.P. and M.L.; experiment, J.L., F.W. and L.L.; writing-original draft preparation, L.L. and J.P.; writing—review and editing, J.P. and M.L.; supervision, J.P. All authors have read and agreed to the published version of the manuscript.

**Funding:** This research was funded by the Key Realm R and D Program of Guangzhou, grant number 202007030005, the Guangdong Natural Science Foundation under grant 2019A1515011375; and the National Natural Science Foundation of China under grants 61876067 and 61906019.

**Data Availability Statement:** The data presented in this study are available on request from the corresponding author.

**Conflicts of Interest:** None of the authors have potential conflict of interest to be disclosed.

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
