# Peer review of "License Plate Image Reconstruction Based on Generative Adversarial Networks"

_remotesensing, doi:10.3390/rs13153018_

Round 1
Reviewer 1 Report
This paper is worth for acceptance, novelty of the idea seems interesting and small changes need to be incorporated in order to enhance.
This paper deals with an exciting topic. The article has been read carefully, and some crucial issues have been highlighted in order to be considered by the author(s).
All the acronyms should be defined and explained first before using them such that they become evident for the readers.
The paper needs to be restructured in order to be precise. (Adding Conclusion section)
The Introduction and related work parts give valuable information for the readers as well as researchers. In addition recent papers should be added in the part of related work.
Grammatical errors should be validated.
It would be good if security domains, such as backdoor [1], would be reflected in future research or related work.
[1] Kwon, Hyun, Hyunsoo Yoon, and Ki-Woong Park. "Multi-targeted backdoor: Indentifying backdoor attack for multiple deep neural networks." IEICE Transactions on Information and Systems 103.4 (2020): 883-887.
Reviewer 2 Report
The paper presents a novel Generative adversarial model setup for License Plate Image Reconstruction with four components that report higher performance and brings a very useful contribution to the field.
Here are some of my comments.
- line 39 has a typo
- The introduction section needs more references to the state-of-the-art methods and possible applications of the proposed approach. Also, a rationale for why GAN is chosen as the main approach can be elaborated with proper reference and evidence. An overall figure of the proposed method could be added here together with some striking results.
- In the figures, the overuse of blue color in the background is making it hard to read.
- A type on page 177
- Section 3.3 needs more elaboration
- Are those codes will be made available for the research community?
